# Post-Acquisition Changes in Agency Cost of Acquirers: Effect of Target Companies

**Prateek Nanda \* and Arun Kumar Gopalaswamy**

Department of Management Studies, Indian Institute of Technology Madras, Chennai 600036, India; garun@iitm.ac.in
* Correspondence: ms18d011@smail.iitm.ac.in

**Abstract:** Acquisitions constitute substantial corporate investments, often leading to changes in ownership and top management giving rise to possible conflicts of interest. The impacts of such conflicts following an acquisition are absorbed by the acquirer and are referred to as agency costs. This study focuses on exploring the influence of the target companies on changes in the post-acquisition agency costs of acquiring companies. A panel fixed effects model is used to analyze acquisitions that took place between 2008–09 and 2019–20. The study's findings indicate that post-acquisition changes in the agency costs of acquirers significantly vary based on the presence of domestic and foreign promoters in the target company. Further promoter groups such as domestic promoters and foreign promoters contribute to conflicting interests, exacerbating post-acquisition agency costs. The monitoring role assumed by foreign promoters of target companies plays a pivotal part in reducing the post-acquisition agency costs of acquirers. Foreign promoters also positively influence post-acquisition profitability by adversely affecting operating expenses, suggesting that they mitigate agency costs by exerting control over management through the monitoring of debt, cash, and profitability. The post-acquisition utilization of the target's cash reserves positively correlates with the operating expenses of the acquirer. It is observed that the acquisition of larger targets magnifies agency costs.

**Keywords:** agency cost; conflict of interest; mergers & acquisitions; firms performance

## 1. Introduction

The primary objective of this study is to investigate the influence exerted by target companies on post-acquisition variations in the agency costs incurred by the acquiring company. During the acquisition process, acquiring companies encounter agency costs associated with the target companies. To illustrate, in May 2013, Apollo Tyres Ltd. entered into an agreement to acquire Cooper Tire & Rubber Co., a U.S.-based company, for approximately $2.5 billion. This transaction aimed to position Apollo Tyres as the world's seventh-largest tire manufacturer. However, Cooper Tire failed to furnish financial statements for its largest subsidiary in China, Chengshan Tyre Co., and grappled with issues concerning its steelworkers' union, which held authority over the approval of the change in ownership. Part of the deal entailed the purchase of a 35% stake in Chengshan Tyre Co., yet the parties involved could not reach a consensus regarding the valuation of this stake. Consequently, Apollo Tyres could not proceed with the acquisition until these issues were resolved. The dispute between Cooper and its subsidiary eventually became entangled in local courts, leading to the abandonment of the deal. Apollo Tyres incurred a breakup fee of $112 million, whereas Cooper was liable for a fee of $50 million (The Economic Times 2014). This scenario underscores the direct impact of the agency costs of the target company on the acquiring entity.

The significance of the target company's agency costs is underscored by three key considerations. Firstly, conflicts between managers and shareholders are highly sensitive to

the investment decisions of firms. The investment decisions made by acquirer management can skew toward excessive conservatism or unwarranted risk-taking, thereby impacting shareholder wealth. Consequently, acquisitions that represent substantial investments, such as the acquisition of a target company, have the potential to influence post-acquisition agency costs.

Secondly, when acquiring a non-controlling stake in the target company, the acquirer becomes a minority shareholder in the target. This places the acquirer in a vulnerable position, susceptible to mismanagement and oppression by the controlling shareholders of the target company, which, in turn, affects the post-acquisition agency costs incurred by the acquiring entity.

Thirdly, acquisitions are frequently characterized by high levels of information asymmetry, which, in turn, affects the post-acquisition agency costs of the acquiring entity. Throughout the acquisition process, acquirers make payments as part of the purchase consideration, often relying on information provided by the target companies. This information serves as the basis for finalizing the deal and determining the purchase consideration. However, due to information asymmetry, the management and owners of the target company can act in a self-serving manner, thereby influencing the post-acquisition agency costs of the acquiring entity. For example, in November 2012, Hewlett Packard (HP) had to write off $8.8 billion of the $11.7 billion it paid to acquire Autonomy Corporation plc, a technology company listed on the FTSE 100. This write-off resulted from accounting manipulation and cover-up by Autonomy, leading HP's shareholders to file a class-action suit against HP's management (The Guardian 2013).

In this paper, the examination of the impact of the target company on post-acquisition changes in agency costs is pursued by the formulation of three sub-objectives. The first sub-objective aims to investigate the influence of the target company's financial aspects on post-acquisition changes in agency costs. For instance, this entails an analysis of how acquirers allocate cash reserves for target companies and the way the target's debt holdings affect the agency costs incurred by the acquirer following the acquisition.

The second sub-objective is to explore the role of target owners, including institutional shareholders of the target, domestic promoters, and foreign promoters, in shaping post-acquisition alterations in agency costs. The acquisition process results in a complete/partial transfer of control for the target company's shareholders. Controlling shareholders, such as domestic promoters, enjoy private control-related benefits and may negotiate for a higher premium in exchange for a reduced stake.

Lastly, foreign promoters of the target company seek to reduce information asymmetry between themselves and domestic promoters by engaging in monitoring activities that involve overseeing the management. This monitoring often takes the form of financial measures such as increased debt and cash monitoring. Therefore, the third sub-objective is to gain an understanding of the mechanisms through which foreign promoters attempt to exercise control.

Agency costs of controlling shareholders are estimated to be 6–25% of a firm's value (Cronqvist and Nilsson 2003). During mergers such costs come to the surface and these agency costs are examined in this paper using a panel fixed effects model. The findings suggest that foreign promoters mitigate such costs whereas domestic promoters exacerbate such costs.

The rest of this paper is structured as follows. Section 2 presents literature review and hypothesis development. Section 3 presents how the data were collected, cleaned, and organized, i.e., data and sample. Section 4 presents the methodology and Section 5 presents the results and discussion. Lastly, in Section 6, the conclusion of this study is presented.

## 2. Literature Review and Hypothesis Development

This section explores the impact of target companies on post-acquisition changes in the agency cost of acquirers, referencing several key papers in the field. However, there are significant gaps in the existing literature. Most studies within the M&A domain focus

on deals where ample data on both acquirer and target are available, often leading to the unintentional exclusion of deals and potentially misleading research outcomes (Netter et al. 2011)[1]. Despite the extensive body of M&A literature, there is a lack of exploration regarding the variables that might influence the post-acquisition performance of acquirers (King et al. 2004). The emphasis in M&A studies tends to skew toward acquiring companies, neglecting a comprehensive examination of the role played by target companies (Yaghoubi et al. 2016b).

Building on the review of agency cost by Panda and Leepsa (2017), it is noteworthy that variables used to proxy agency cost and those measuring post-acquisition performance bear similarities but differ in their interpretations. For example, metrics like asset utilization ratio (ATO) and operating expense ratio (OER) often serve as proxies for agency cost. However, while ATO measures managerial efficiency in asset utilization (higher ATO implying lower agency cost), OER gauges managerial control over operating expenses (lower OER indicating lower agency cost). Interestingly, these accounting ratios are also employed to measure the post-acquisition performance of acquirers, as indicated by King et al. (2004). While many M&A studies traditionally examine acquisition performance using market measures like BHAR (buy and hold abnormal returns), CAR (cumulative announcement returns), and Tobin's Q, relying solely on market-based measures can lead to misspecification due to increased variance associated with the event of acquisitions (Kothari and Warner 2004). Hence, this study opts to use accounting measures as a proxy to gauge agency cost.

According to Yaghoubi et al. (2016a), a target's characteristics are a key determinant of acquisition performance. Prior studies primarily examined two such target characteristics, the nature of the target company (public/private) and the pre-merger performance of the target company. For instance, Moeller et al. (2005) argued that the acquisition of private targets in the USA from 1980 to 2004 generated higher announcement returns compared to the acquisition of public targets during the same period. Similar findings are reported by other researchers, such as Chang (1998), Fuller et al. (2002), Conn et al. (2005), and Faccio et al. (2006). Similarly, the pre-merger performance of the target is one of the characteristics that affect the post-acquisition returns of acquirers. For instance, Lang et al. (1989) and Servaes (1991) posit that acquisition abnormal returns are larger when targets have low Tobin's Q ratios. Morck et al. (1990) find that announcement abnormal returns to the bidders are negatively correlated with the pre-announcement performance of the targets in non-banking industries. The meta-analytic review by King et al. (2004) also argues that shareholders of target companies gain abnormal market returns upon acquisition announcement whereas returns of acquiring companies are insignificant or negative. This is because expected synergies are not realized subsequently, as observed from accounting returns, i.e., ROA (return on assets), ROE (return on equity), and ROS (return on sales). The implication of these findings is that the investors of the target company if given a choice to either cash out or receive an equity stake, should cash out.

Some studies have examined the relationship between target returns and target ownership and suggest that there exists a positive relationship between inside ownership and target returns (Bauguess et al. 2009). This study adds depth to the existing literature by examining how target ownership impacts acquirers. It is plausible that controlling shareholders of target companies derive private benefits of control, and an acquisition might potentially dilute their stake, leading to a potential relinquishment of these private benefits. As a result, controlling shareholders of target companies often negotiate for a higher premium in exchange for a lower stake, allowing them to retain substantial control. Acquirers frequently acquiesce to these negotiations, opting for a non-controlling stake. This strategic move grants them access to valuable resources such as the target's distribution channels, technological innovations, and preferential pricing for goods or services (Bogert 1996; Dushnitsky and Lenox 2006). However, this decision also exposes acquirers to the agency issues of the controlling shareholders of the target company. By accepting a non-controlling stake, acquirers gain certain strategic advantages but also inherit potential challenges associated

with the controlling shareholders of the target company. This dynamic represents a delicate balance between accessing valuable resources and navigating the agency problems that may arise from the target's controlling shareholders, thereby providing the empirical motivation for this study.

**Hypothesis 1 (H1).** *Domestic promoters of target companies exacerbate the post-acquisition agency cost of acquirers.*

Unlike, their domestic counterparts, foreign promoters often exhibit a distinct set of expectations and demands when involved in acquisitions. Foreign promoters typically advocate for enhanced transparency through increased disclosure, improved accounting and auditing standards, the utilization of international auditors, aligned incentives, a longer-term investment outlook, and more extensive monitoring protocols (Choi et al. 2013). In the post-acquisition phase, the involvement of foreign promoters within target companies tends to facilitate a smoother integration process with acquirers. This active involvement and support from foreign promoters contribute significantly to mitigating the post-acquisition agency costs incurred by acquirers. Their emphasis on improved integration strategies aids in reducing the challenges typically associated with post-acquisition transitions, ultimately benefiting the acquirer by lowering agency costs.

However, foreign ownership is also associated with a short-term investment horizon, superior information processing capability, and controlling positions, which can create conflicts of interest with other shareholders. Therefore, this study contributes to the literature on the impact of a target's foreign ownership.

**Hypothesis 2 (H2).** *Foreign promoters of target companies would lower the post-acquisition agency cost of acquirers.*

Post-acquisition, acquirers wield a certain degree of influence over the assets of the acquired companies. This influence can be strategically utilized to address potential agency problems. For instance, there's a longstanding argument that the utilization of debt holdings can effectively mitigate conflicts of interest between managers and shareholders in acquiring companies, particularly those grappling with free cash flow issues. Huang et al. (2018) assert that the presence of debt can serve as a mechanism to alleviate such conflicts. Additionally, they highlight that short-maturity debt can assist cash-rich acquirers in striking more favorable deals, resulting in higher announcement returns and improved post-acquisition operational performance.

Extending the findings of Huang et al. (2018), which examined the role of the acquirer's debt, this study aims to delve deeper into the role played by the target's debt holdings, cash reserves, and profitability in explaining changes in the acquirer's agency costs. For instance, high cash reserves within target companies could be utilized for either precautionary motives or, potentially, driven by agency considerations (Gao and Mohamed 2018). This study seeks to explore how these financial aspects of the target company, such as debt holdings, cash reserves, and profitability, could impact the agency costs incurred by the acquirer post-acquisition. By investigating the interplay between these factors, the study aims to offer a more nuanced understanding of how financial dynamics within the target company sphere can influence the agency costs of acquirers.

**Hypothesis 3 (H3).** *The financials of the target company would affect the post-acquisition agency cost of acquirers.*

Foreign promoters often encounter a common challenge of information asymmetry during acquisition (Choi et al. 2013). Post-acquisition, foreign promoters act to quickly reduce this asymmetry by enhancing their monitoring mechanisms. It can be argued that foreign promoters adopt a strategy such as increasing debt to ensure discipline or aggressively plan for sales growth post-acquisition for better control and information flow.

Foreign promoters wield the ability to fulfill a monitoring role within a company through their influence on its assets and liabilities. For instance, as highlighted by Huang et al. (2018), foreign promoters have the capacity to utilize strategic measures to mitigate managerial tendencies towards overspending. One such mechanism involves the manipulation of debt holdings. By increasing debt holdings within the company, foreign promoters can effectively discipline managers and reduce their inclination to overspend. This strategy works as a form of checks and balances, exerting pressure on managers to operate within constraints, thereby curbing potential tendencies towards excessive spending. This disciplined approach is facilitated by the increased obligation that comes with higher debt, leading managers to be more cautious and strategic in their financial decisions.

**Hypothesis 4 (H4).** *Foreign promoters of target companies would lower the post-acquisition agency cost of acquirers by exerting influence through the financials of the target company.*

This study aims to extend the existing literature by focusing on the often-neglected influence of target companies on post-acquisition agency cost, contributing to a more comprehensive understanding of M&A dynamics.

## 3. Data and Sample

This study represents a comprehensive exploration of the influence of Indian target companies on post-acquisition adjustments in the agency cost borne by acquirers. The empirical data utilized in this inquiry was meticulously acquired from the Centre for Monitoring Indian Economy (CMIE). To ensure the study's methodological soundness and practical applicability, a deliberate approach was taken in selecting the sample of acquirers. Consequently, certain categories of acquirers, including financial institutions, non-banking financial institutions, banks, mutual funds, and insurance companies, were deliberately excluded from the sample. This decision was grounded in the recognition of the highly regulated nature of these entities, which could introduce unique dynamics that might confound the analysis. Furthermore, transactions that obtained approval from directors (subsequently discarded), preferential allotments, acquisitions by private firms, and those involving high-net-worth individuals were carefully excluded from the scope of this study. Additionally, transactions featuring multiple acquirers were consciously set aside from the sample. This meticulous curation was carried out to maintain the primary research focus, which centers on the scrutiny of post-acquisition agency cost changes for acquiring companies. The inclusion of transactions with multiple acquirers could introduce complexity and potential bias into the analysis. Moreover, transactions in which the financial data of the target companies were conspicuously absent from the CMIE dataset were judiciously omitted from consideration. Following this rigorous data refinement process, a total of 91 transactions were retained for comprehensive analysis, each of which provided a comprehensive set of financial data pertaining to the target companies.

The study is carried out in the Indian context for two reasons. Firstly, in India, a large[2] percentage of businesses are family-owned, leading to concentrated ownership unlike in other high-income nations (Sarkar and Sarkar 2000). Secondly, the coexistence of concentrated ownership in a weak corporate governance regime might reduce the stress on shareholder-manager conflict but can induce shareholder–shareholder conflict (Villalonga and Amit 2006). In India, the average value of the "rule of law" index is 4.9, which is notably lower than the average of 9.6 in countries such as the USA, UK, Germany, Japan, Australia, and Canada (La Porta et al. 1998).

The study is based on deals that took place from 2008–09 to 2019–20. This sample period was chosen as there were significant regulatory changes mandated for companies in India after the 2007 financial crisis. One notable instance was SEBI's amendment of clause 49 of the listing agreement through a circular issued on 8 April 2008, which mandated the inclusion of independent directors in the company board of up to 50% of total board seats. Furthermore, a study by UNCTD[3]. emphasized the importance of improving board

capacity in emerging markets in the aftermath of the financial crisis, further justifying the relevance of the chosen sample period. Additionally, the selection of deals prior to this period may skew the results. The change in agency cost would be measured for three financial years after acquisition because post-acquisition integration could be as long as three years; therefore, the last year for our sample is restricted to 2019–20.

To mitigate the potential impact of industry-specific effects that could unduly sway the research findings, the target companies were systematically classified based on their 3-digit NIC Codes (National Industrial Classification). A scrutiny of Figure 1 reveals a conspicuous dispersion of target companies across various industries. This distribution effectively dispels any presumption of concentration within a limited set of sectors, thereby enhancing the robustness and generalizability of the study's findings.

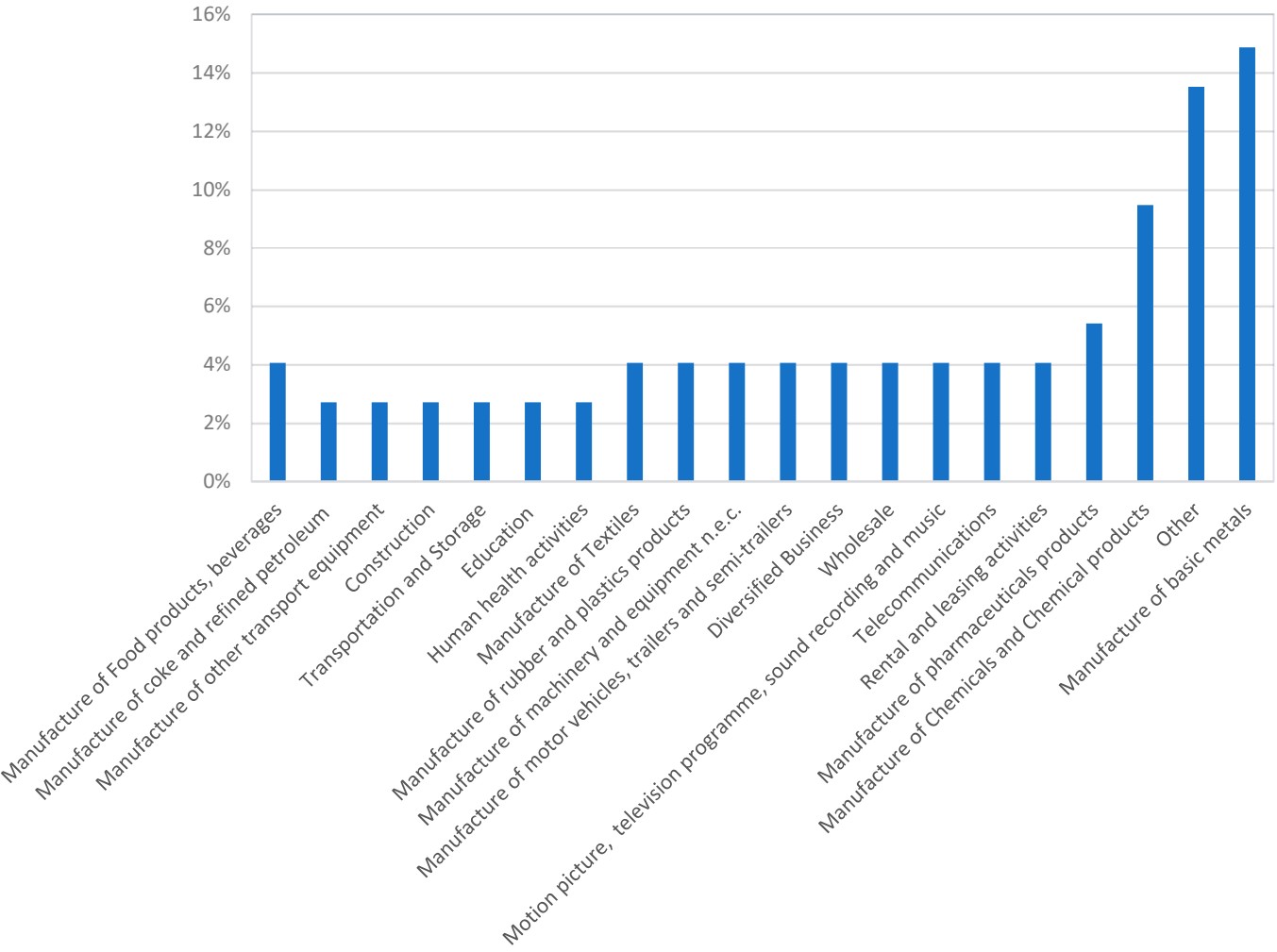

**Figure 1.** Industry Classification of Target Companies.

## 4. Methodology

As the dataset is structured in a panel format, it is necessary to determine whether a fixed effects or random effects model is more suitable for analysis. To make this determination, the Hausman specification test (Hausman 1978) was employed. This test assesses strict exogeneity, and its outcome indicates whether the preferred model is a random effects model in the absence of a correlation or a fixed effects model when a correlation is present. In this case, the Hausman specification test yielded a $p$-value of less than 0.01, suggesting that the characteristics of the target company do not exhibit a normal distribution. Consequently, a fixed effects model was selected for this study.

The study utilizes unbalanced panel data, which permits the capture of changes in the agency cost of target companies over time, specifically in response to acquisition events. The model employed in this study is as follows:

$$\Delta AC_{it} = \alpha_1 + \beta_1 \, Tar\_Prom_{it} + \beta_2 \, Tar\_Inst_{it} + \beta_3 \, F\_Dummy_{it} + \beta_4 \, Tar\_Cash_{it} + \beta_5 \, Tar\_Lev_{it} + \beta_6 \, Tar\_PBDITA_{it} + \beta_7 \, Rel\_Size_{it} + \beta_8 (Interaction \; terms)_{it} \; U_{it} \tag{1}$$

### 4.1. Dependent Variables

All the variables utilized in this study are comprehensively detailed in Table 1. Change in agency cost would be measured using "change" in asset utilization (ATO) and Operating Expense Ratio (OER), which is consistent with the prior literature (Ang et al. 2000; Singh and Davidson 2003; Fleming et al. 2005; Florackis and Ozkan 2009; Rashid 2013; Golubov and Xiong 2020).

**Table 1.** Description of Variables used in this study.

| Sl.no | Name of the Variable | Explanation |
|---|---|---|
| **Variables Measuring Agency Cost of Acquirer** | | |
| 1 | Delta ATO | Difference between acquirer's ATO for post-acquisition financial years with the financial year ending prior to acquisition. |
| 2 | Delta OER | Difference between acquirer's OER for post-acquisition financial years with the financial year ending prior to acquisition. |
| 3 | Delta Cash | Difference between acquirer's cash reserves (scaled to total assets) for post-acquisition financial years with the financial year ending prior to acquisition |
| **Target Ownership Variables** | | |
| 4 | Tar_Prom | Percentage of equity holdings owned by domestic promoters. |
| 5 | Tar_Inst | Percentage of equity holdings owned by institutional promoters. |
| 6 | F_Dummy | A binary variable that takes the value 1 when foreign owners are present in the target company and 0 otherwise. |
| **Target Financial Variables** | | |
| 7 | Tar_Cash | Target company's cash reserve scaled to total assets |
| 8 | Tar_PBDITA | Target company's profit before depreciation interest tax and amortization scaled to total assets |
| 9 | Tar_Lev | Target's debt scaled to total assets |
| 10 | Rel_Size | Target's book value of assets/Acquirer's book value of assets |
| **Interaction terms and others** | | |
| 11 | F_dumXT_Cash | Interaction variable between F_Dummy and Delta Cash. |
| 12 | F_DumXT_Debt | Interaction variable between F_Dummy and Delta Debt. |
| 13 | F_DumXT_PBDITA | Interaction variable between F_Dummy and PBDITA. |

An argument against ATO as the proxy is that higher sales may not lead to an increase in shareholder wealth because sales may not be from profitable avenues. Sales from operating activities may be expropriated by management and would not be distributed to shareholders. Coles et al. (2006) argue that ATO greatly varies even between companies operating in the same industry. Furthermore, an argument against the use of ATO in this study could stem from the impact of merger accounting. When mergers occur, goodwill is recorded in the books of the acquirer, leading to an increase in the total assets, i.e., the denominator in the ATO calculation. As a result, the post-acquisition ATO may appear lower, potentially raising concerns that ATO might not effectively measure agency cost and could be influenced by accounting adjustments.

However, proponents of using ATO argue that it still provides valuable insights into agency cost, i.e., higher ATOs may be indicative of efficient management and effective resource utilization. For instance, acquirers with lower agency costs, such as private acquirers characterized by a lower separation between management and owners, tend to exhibit higher ATOs (Golubov and Xiong 2020) compared to public acquirers. Similarly, Bargeron et al. (2008) concluded that private acquirers pay lower acquisition premiums than public acquirers, indicating that private acquirers may be more cost-conscious and vigilant in their acquisition decisions, further supporting the association between lower agency costs and higher ATOs. Further, ATO captures the loss in revenue per dollar of investments that could be attributable to inefficient asset utilization (Ang et al. 2000; Singh and Davidson 2003).

Following Ang et al. (2000), operating expense ratio is used to complement the use of ATO as a proxy for agency cost. This ratio captures how effectively the management of acquirers controls operating expenses. Unlike ATO, this measure of agency cost is a direct proxy for agency cost.

Jensen's (1986) free cash flow hypothesis argues that excess cash can be misused by managers as it provides an incentive to the managers to waste excess cash on unprofitable acquisitions. The literature has used cash as proxies to measure agency costs (Song et al. 2007; Yen and André 2007). In this study, changes in cash reserves are used to measure "change" in agency cost of acquirers.

### 4.2. Independent Variables

4.2.1. Target Ownership Variables

The ownership structure of the target company can potentially exacerbate the principal–principal conflicts that arise following an acquisition. Therefore, this study incorporates the ownership by the target company's domestic promoters, institutional shareholders, and foreign promoters.

In the context of acquisitions, agency conflicts often intensify due to the presence of a controlling shareholder in the target company. This controlling shareholder typically seeks to negotiate a higher purchase consideration while offering a lower stake, thus retaining control. In cases where the stake acquired is non-controlling, the acquirer becomes a minority shareholder in the target company. In such scenarios, the target's controlling shareholder and managers may influence the target's decisions in their favor, potentially expropriating the acquirer, as theorized by Jensen and Meckling (1976) and Johnson et al. (2000). Conversely, acquirers may opt for a minority stake in pursuit of other objectives, such as joint product development, improving information flows, and relationship-specific investments, as suggested by Allen and Phillips (2000), Fee et al. (2006), and Ouimet (2013). Consequently, the impact of the target company's owners on post-acquisition changes in agency costs remains uncertain.

Regarding foreign promoters, their presence is hypothesized to have two conflicting effects on information asymmetry. On the one hand, they are often associated with shorter investment horizons, information-sharing challenges, and increased conflicts with other shareholders, which can contribute to higher agency costs. On the other hand, foreign promoters may drive demand for improved disclosure, effective monitoring, and higher standards of accounting and auditing, which could lead to a reduction in agency costs, as observed in the findings of Choi et al. (2013). Institutional shareholders often hold stakes in both acquirer and target companies. Therefore, they are concerned with overall value creation. This exacerbates conflicts of interest between the shareholders of target and acquirer companies. Conflicts of interest are also affected because the dollar value of investment by institutional shareholders doesn't change but their degree of control and ownership would change after acquisition.

4.2.2. Target Financial Variables

After an acquisition, acquirers have a certain claim on a target's assets and liabilities which has a bearing on post-acquisition agency cost. Therefore, the financials of the target company are used in this study, i.e., the target's cash reserves, the target's profitability, the target's leverage, and the relative size of the target with respect to the acquirer.

The target's cash reserves represent a vital part of the company's assets but they have their costs. These cash reserves may be used by the acquirer to smoothen the integration with the target or could be used by the acquirer in an agency-driven manner (Harford 1999).

The target's leverage might lower the agency conflicts within the target. However, the impact of such leverage after acquisition is quite uncertain. On the one hand, if the target's size is relatively small compared to the acquirer, then the impact of debt may be very nominal; however, if the size is significant, then this affects the post-acquisition agency cost of the acquirer. For instance, if a company buys a non-controlling stake in a larger company, then the impact of the target's leverage will be significant.

The target's profitability is quite debated in the extant literature. Target companies having poor profitability may be interpreted as firms with a high degree of agency conflicts whereas target companies with higher profitability are characterized by a low degree of agency conflicts. When these companies are integrated with the acquirer, this should affect the acquirer's post-acquisition agency cost.

## 5. Results and Discussion

### 5.1. Descriptive Results

Table 2 presents the descriptive statistics of all variables. From the table, it can be observed that asset utilization of the acquirer companies has decreased on average by 7.99%. Whereas, the operating expense ratio decreased by 14.2%. Overall, this indicates that acquisitions in our sample are aimed at reducing costs. The acquirer's cash reserves fall by 1.3% on average. This could be due to increased post-integration cash requirements.

**Table 2.** Descriptive Statistics.

| Variable | Obs | Mean | Std. Dev. | P25 | P75 |
|---|---|---|---|---|---|
| Delta ATO | 364 | −0.0799 | 0.228 | −0.188 | 0.038 |
| Delta OER | 364 | −0.142 | 0.427 | −0.419 | 0.10 |
| Delta Cash | 364 | −0.013 | 0.039 | −0.019 | 0.005 |
| Tar Prom | 358 | 52.788 | 20.752 | 39.19 | 69.92 |
| F Dummy | 364 | 0.146 | 0.353 | 0 | 0 |
| Tar Inst | 338 | 8.842 | 13.13 | 0.18 | 13.79 |
| Tar Cash | 364 | 0.027 | 0.041 | 0.002 | 0.03 |
| Tar PBDITA | 364 | 0.081 | 0.088 | 0.014 | 0.14 |
| Tar Lev | 364 | 0.154 | 0.244 | 0 | 0.212 |
| relativesize | 364 | 1.391 | 4.398 | 0.066 | 1.021 |

Descriptive statistics of Indian target companies over the period of 2008–2019. Data are collected from CMIE. The net number of acquisitions is 91. Financial year before acquisition and post-acquisition analysis up to 3 years are included in the sample, therefore, the total number of observations is 364 (91 × 4). Variables are winsorized at 5%.

On average, domestic promoters of the target company hold a controlling stake of 52.788%. The average institutional ownership of target companies is 8.842%. The target's cash reserves represent 2.7% of their total assets. The target's PBDITA is 8.1%. The target's debt holdings represent 15.4% of their total liabilities.

Table 3 presents statistical evidence of differences in changes in agency cost based on target promoters. The results of the Wilcoxon–Mann–Whitney test reported in panel A of Table 3 suggest that post-acquisition agency cost varies significantly for acquirers who acquired targets having a controlling shareholder as against targets who do not hold a controlling shareholder. The results in Panel B of Table 3 suggest that post-acquisition agency cost varies significantly between targets who have a foreign promoter and targets who do not have a foreign promoter.

**Table 3.** Change in agency cost based on target's promoters.

| Panel A | Target Domestic Promoter's Holding a Controlling Stake | | | |
|---|---|---|---|---|
| **Proxy for Agency Cost** | **Controlling Target Promoter Exists (Mean)** | **Controlling Target Promoter Does Not Exist (Mean)** | **Wilcoxon–Mann–Whitney (*p*-Value)** | **Median** |
| Delta ATO | −0.055 | −0.088 | 0.0556 * | −0.029 |
| Delta OER | −0.172 | −0.043 | 0.158 | −0.081 |
| **Panel B** | **Presence of Target Foreign Promoters** | | | |
| **Proxy for Agency Cost** | **Foreign Target Promoter Exists (Mean)** | **Foreign Target Promoter Does Not Exist (Mean)** | **Wilcoxon–Mann–Whitney (*p*-Value)** | **Median** |
| Delta ATO | −0.1528 | −0.054 | 0.0195 ** | −0.029 |
| Delta OER | −0.0301 | −0.136 | 0.439 | −0.081 |

**, * denotes rejection of null hypothesis at 5%, and 10% significance levels, respectively.

Table 4 presents the results of the pairwise correlation of all the variables. It can be observed that the acquirer's change in cash reserves is positively associated with the acquirer's Delta ATO, indicating that post-acquisition, cash utilization is for precautionary motive and not agency-driven. The target's promoter holdings are negatively associated with the acquirer's Delta OER, suggesting that target promoters try to reduce the post-acquisition operating expense of acquirers. This could be attributed to the fact that acquisitions present in the sample are characterized by the acquisition of companies that have a lower agency cost, resulting in an overall reduction in the post-acquisition agency cost of acquirers. However, the presence of foreign promoters in the target is negatively associated with a change in the acquirer's Delta ATO. This is because the controlling stake of domestic promoters usually enables them to extract private benefits of control and foreign promoters of the target company employ their monitoring mechanism to lower such rent-extracting behavior. This is evident from the negative and significant correlation between foreign promoters and domestic promoters. Further, target companies also experience shareholder–shareholder conflict between institutional shareholders and domestic promoters, which is evident from the negative and significant association between the two. This conflict of interest between different shareholders results in a negative association with Delta Cash. The target's institutional shareholders have a positive and significant relation with the target's profitability.

**Table 4.** Pairwise Correlation.

| Variables | (1) | (2) | (3) | (4) | (5) | (6) | (7) | (8) | (9) | (10) |
|---|---|---|---|---|---|---|---|---|---|---|
| (1) Delta_ATO | 1.000 | | | | | | | | | |
| (2) Delta_OER | 0.070 | 1.000 | | | | | | | | |
| (3) Delta_Cash | 0.119 ** | 0.031 | 1.000 | | | | | | | |
| (4) Tar_Prom | 0.061 | −0.091 * | −0.056 | 1.000 | | | | | | |
| (5) F_Dummy | −0.107 ** | 0.052 | −0.162 *** | −0.245 *** | 1.000 | | | | | |
| (6) Tar_Inst | −0.012 | 0.042 | −0.101 * | −0.330 *** | 0.036 | 1.000 | | | | |
| (7) Tar_Cash | 0.057 | 0.033 | 0.104 ** | 0.171 *** | −0.043 | 0.015 | 1.000 | | | |
| (8) Tar_PBDITA | −0.046 | 0.045 | −0.066 | 0.014 | 0.052 | 0.206 *** | 0.107 ** | 1.000 | | |
| (9) Tar_Lev | 0.041 | −0.046 | 0.070 | 0.187 *** | −0.057 | 0.002 | −0.116 ** | −0.193 *** | 1.000 | |
| (10) relativesize | 0.056 | 0.132 ** | −0.299 *** | −0.031 | 0.193 *** | 0.180 *** | −0.076 | 0.035 | 0.000 | 1.000 |

***, **, * denotes rejection of null hypothesis at 1%, 5%, and 10% significance levels, respectively.

*5.2. Empirical Findings*

The findings from the regression analysis[4] examining the impact of the target company on the post-acquisition change in agency cost of the acquirer are presented in Tables 5 and 6, which use Delta ATO and Delta OER as proxies for agency cost, respectively.

Findings from Column 1 of Table 5 indicate that the target's cash positively and significantly affects the acquirer's Delta ATO with a coefficient of 0.582. This indicates that the target's cash reserves are utilized for precautionary motives rather than agency motives.

Such findings confirm H3, i.e., the financials of the target company have a bearing on the post-acquisition agency cost of acquirers.

**Table 5.** Panel regression analysis on examining the impact of target company on acquirer's agency cost using Delta ATO as a proxy for agency cost.

| | (1) | (2) | (3) | (4) | (5) | (6) |
|---|---|---|---|---|---|---|
| | Delta_ATO | Delta_ATO | Delta_ATO | Delta_ATO | Delta_ATO | Delta_ATO |
| Tar_Cash | 0.582 ** | 0.211 | 0.203 | 0.021 | 0.197 | 0.201 |
| | (0.263) | (0.321) | (0.318) | (0.356) | (0.315) | (0.318) |
| Tar_PBDITA | −0.14 | −0.066 | −0.049 | −0.05 | −0.069 | −0.096 |
| | (0.114) | (0.113) | (0.112) | (0.112) | (0.111) | (0.124) |
| Tar_Lev | 0.013 | 0.035 | 0.074 | 0.075 | −0.016 | 0.06 |
| | (0.097) | (0.104) | (0.105) | (0.105) | (0.111) | (0.106) |
| Rel_Size | −0.038 *** | −0.039 *** | −0.039 *** | −0.039 *** | −0.038 *** | −0.039 *** |
| | (0.008) | (0.008) | (0.008) | (0.008) | (0.008) | (0.008) |
| Tar_Prom | | −0.002 * | −0.002 ** | −0.003 ** | −0.002 ** | −0.002 ** |
| | | (0.001) | (0.001) | (0.001) | (0.001) | (0.001) |
| Tar_Inst | | −0.003 | −0.003 | −0.003 | −0.003 | −0.004 |
| | | (0.002) | (0.002) | (0.002) | (0.002) | (0.002) |
| F_Dummy | | | −0.195 ** | −0.23 *** | −0.266 *** | −0.236 ** |
| | | | (0.081) | (0.087) | (0.086) | (0.093) |
| F_dumXT_Cash | | | | 0.793 | | |
| | | | | (0.701) | | |
| F_dumXT_Lev | | | | | 0.595 ** | |
| | | | | | (0.26) | |
| F_dumXT_PBDITA | | | | | | 0.258 |
| | | | | | | (0.295) |
| _cons | −0.028 | 0.108 | 0.161 ** | 0.172 ** | 0.169 ** | 0.17 ** |
| | (0.026) | (0.068) | (0.071) | (0.072) | (0.07) | (0.072) |
| Observations | 364 | 332 | 332 | 332 | 332 | 332 |
| R-squared | 0.089 | 0.113 | 0.134 | 0.139 | 0.153 | 0.137 |

Standard errors in parentheses. ***, **, * denotes rejection of null hypothesis at 1%, 5% and 10% significance levels respectively.

**Table 6.** Panel regression analysis on examining the impact of target company on acquirer's agency cost using Delta OER as a proxy for agency cost.

| | (1) | (2) | (3) | (4) | (5) | (6) |
|---|---|---|---|---|---|---|
| | Delta_OER | Delta_OER | Delta_OER | Delta_OER | Delta_OER | Delta_OER |
| Tar_Cash | 0.414 | 1.249 * | 1.262 * | 2.082 *** | 1.257 * | 1.275 * |
| | (0.781) | (0.703) | (0.699) | (0.778) | (0.7) | (0.697) |
| Tar_PBDITA | 0.12 | 0.015 | −0.015 | −0.012 | −0.032 | 0.187 |
| | (0.34) | (0.248) | (0.247) | (0.244) | (0.248) | (0.272) |
| Tar_Lev | −0.588 ** | −0.394 * | −0.462 ** | −0.469 ** | −0.54 ** | −0.403 * |
| | (0.288) | (0.229) | (0.23) | (0.228) | (0.246) | (0.232) |
| Rel_Size | 0.017 | −0.25 | −0.318 | −0.338 | −0.308 | −0.275 |
| | (0.04) | (4.437) | (4.411) | (4.371) | (4.413) | (4.393) |
| Tar_Prom | | −0.006 ** | −0.005 * | −0.004 * | −0.005 * | −0.005 ** |
| | | (0.002) | (0.002) | (0.002) | (0.002) | (0.002) |
| Tar_Inst | | −0.003 | −0.003 | −0.004 | −0.003 | −0.002 |
| | | (0.005) | (0.005) | (0.005) | (0.005) | (0.005) |
| F_Dummy | | | 0.346 * | 0.505 *** | 0.284 | 0.523 ** |
| | | | (0.178) | (0.189) | (0.191) | (0.205) |
| F_dumXT_Cash | | | | −3.553 ** | | |
| | | | | (1.532) | | |
| F_dumXT_Lev | | | | | 0.512 | |
| | | | | | (0.577) | |
| F_dumXT_PBDITA | | | | | | −1.115 * |
| | | | | | | (0.645) |
| _cons | −0.079 | 0.57 | 0.569 | 0.55 | 0.562 | 0.476 |
| | (0.091) | (6.195) | (6.159) | (6.103) | (6.162) | (6.134) |
| Observations | 364 | 332 | 332 | 332 | 332 | 332 |
| R-squared | 0.018 | 0.069 | 0.083 | 0.104 | 0.086 | 0.095 |

Standard errors in parentheses. ***, **, * denotes rejection of null hypothesis at 1%, 5% and 10% significance levels respectively.

However, the findings in Column 1 of Table 5 show that the relative size of the target company, i.e., Rel_Size, negatively and significantly affects Delta ATO with an estimated coefficient of −0.038, indicating that acquisition of a larger stake in target companies exacerbates the agency cost more than the acquisition of smaller target companies i.e., a 1% increase in the relative size of target company would mean a 3.8% decline in post-acquisition ATO.

The findings presented in Column 3 of Table 5 indicate that the estimated coefficient of the target promoters, i.e., Tar_Prom, is −0.02 and is also statistically significant. The results suggest that target promoters could experience conflicts of interest with the shareholders of acquirers, thereby negatively affecting Delta ATO. These findings support H1 of this study, i.e., domestic promoters of the target company exacerbate the agency cost of acquiring companies. Additionally, the findings presented in Column 3 of Table 5 depict a negative and significant estimated coefficient of −0.195 for F_Dummy, i.e., foreign promoters. The results indicate that foreign promoters also experience conflicts of interest with acquirers, resulting in a negative impact on Delta ATO. These findings support H2 of this study, i.e., foreign promoters of the target company exacerbate the agency cost of acquiring companies.

To reduce information asymmetry, the target's foreign promoters would try to exercise control and monitor the target's management through various financial channels such as cash reserves, debt, and profitability. The results presented in column 5 of Table 5 indicate that the presence of foreign promoters coupled with the target's leverage has a significant and positive estimated coefficient of 0.595. It is worth noting that the target's leverage by itself was not statistically significant but the target's foreign promoters can use leverage to mitigate the post-acquisition agency cost of the acquirer. These results confirm H4 of this study, which argues that foreign promoters exert influence through the financials of the company to lower the post-acquisition agency cost of acquirers.

The results also indicate that various other unobservable target characteristics, which are captured by a constant term, positively and significantly affect Delta ATO indicating a lowering of post-acquisition agency cost.

The target's institutional shareholders often fail to mitigate agency cost during mergers. The results presented in column 3 of Table 5 indicate a statistically insignificant coefficient. However, this could be due to a reduction in the degree of control of institutional shareholders.

To add robustness to the results, the proxy to measure the change in agency cost was changed to Delta OER. When Delta OER is used as a proxy to measure agency cost, the impact of the target company's cash reserve is positive and significant with a coefficient of 1.262, as is evident in Column 3 of Table 6. The results indicate that the target's cash reserves are used in an agency-driven manner by the acquirers. The results confirm H3 of the study, which states that the financials of the target company have a bearing on the post-acquisition agency cost of acquirers.

However, due to the presence of foreign promoters in the target company, the agency effects of cash are mitigated. This was evident because the impact of F_dumXT_Cash is negative and significant with a coefficient of −3.553, as depicted in Column 4 of Table 6. It is worth noting that when Delta ATO was used as a proxy, the majority of the evidence pertaining to target cash reserves was not statistically significant, as depicted in Table 5. Additionally, when Delta ATO was used as a proxy, the interaction of cash reserves with F_dummy, i.e., F_dumXT_Cash, was not statistically significant (estimated coefficient of 0.793). Thus, it can be argued that the monitoring role due to the presence of foreign promoters lowers post-acquisition operating expenses by monitoring the target's cash reserves, thereby confirming H4 of the study, which states that foreign promoters exert influence through financials, which lowers the post-acquisition agency cost.

However, the target's debt negatively and significantly affects the acquirer's Delta OER with an estimated significant coefficient of −0.462. The results suggest that target companies' debt mitigates the post-acquisition agency cost of acquirers, thereby confirming H3 of the study. It is worth noting that when Delta ATO was used as a proxy, the estimated coefficient for the target's leverage was 0.074 and was statistically insignificant, as depicted in column 3 of Table 5. Conversely, interacting F_Dummy with leverage gives statistically

insignificant results when Delta OER is used as a proxy. Therefore, it can be argued that the target's leverage disciplines the managers and lowers the post-acquisition operating cost. The target's foreign promoters use debt to monitor the management and try to enhance the post-acquisition revenue, thereby reducing post-acquisition agency cost, and thereby, confirming H4 of the study.

The results in Column 3 of Tables 5 and 6 suggest that when agency cost is measured as Delta ATO, F_Dummy is negative and statistically significant with an estimated coefficient of −0.195. However, when agency cost is measured as Delta OER, F_Dummy is positive and significant with an estimated coefficient of 0.346. The results complement each other and indicate that the exercising of their monitoring function by foreign promoters lowers the post-acquisition agency cost of the acquiring company. The results support H2 of the study, which suggests that foreign promoters of the target reduce the post-acquisition agency cost of acquirers.

The results in Column 6 of Table 6 suggest that the exercising of their monitoring role by foreign promoters has a favorable impact on the target company's profitability. The estimated coefficient of F_dumXT_PBDITA is −1.115 and is statistically significant. The results indicate that due to the presence of foreign promoters, profitability increases and post-acquisition operating expenses fall. However, when Delta ATO was used as a proxy, the estimated coefficient of F_dumXT_PBDITA was not statistically significant. The results confirm H4 of the study and indicate that foreign promoters enhance the target's profitability by monitoring the operating cost.

Additional Test

To further the understanding of the impact of target companies on the post-acquisition agency cost of acquirers, this sub-section provides additional evidence on the impact of "pre-merger" target characteristics on the post-acquisition change in agency cost of acquirers. In this section, pre-merger target characteristics are regressed with the 1-year post-acquisition change in cash of acquirers. Delta Cash is used as a proxy to measure the change in agency cost because cash fosters agency behavior.

The findings presented in column 3 of Table 7 indicate that the target's relative size to the acquirer, i.e., Relative_Size, has an estimated significant coefficient of −0.06. The results indicate that the acquisition of larger target companies exacerbates the agency cost of acquirers. The results indicate that a 1% increase in the relative size of the target could result in a 6% decrease in the cash of the acquirer.

**Table 7.** Cross-sectional OLS regression analysis on examining the impact of pre-merger target company on acquirer's agency cost using Delta Cash as a proxy for agency cost.

| | (1) Delta_Cash | (2) Delta_Cash | (3) Delta_Cash | (4) Delta_Cash | (5) Delta_Cash | (6) Delta_Cash |
|---|---|---|---|---|---|---|
| Tar_Cash | 0.012 | 0.064 | 0.067 | 0.061 | 0.063 | 0.066 |
| | (0.075) | (0.091) | (0.09) | (0.091) | (0.09) | (0.091) |
| Tar_PBDITA | −0.006 | 0.001 | 0.007 | 0.011 | 0.003 | 0.013 |
| | (0.06) | (0.06) | (0.059) | (0.06) | (0.059) | (0.072) |
| Tar_Lev | 0.006 | 0.017 | 0.019 | 0.02 | 0.004 | 0.019 |
| | (0.042) | (0.041) | (0.041) | (0.041) | (0.045) | (0.041) |
| Relative_Size | −0.006 *** | −0.007 *** | −0.006 *** | −0.006 *** | −0.006 *** | −0.006 *** |
| | (0.001) | (0.001) | (0.001) | (0.001) | (0.001) | (0.001) |
| Tar_Prom | | −0.0005 | −0.001 ** | −0.001 ** | −0.001 * | −0.001 ** |
| | | (0.0003) | (0.0003) | (0.0003) | (0.0003) | (0.0003) |
| Tar_Inst | | −0.0002 | −0.0004 | −0.0004 | −0.0004 | −0.0004 |
| | | (0.0004) | (0.0004) | (0.0004) | (0.0004) | (0.0004) |
| F_Dummy | | | −0.033 * | −0.041 | −0.043 * | −0.032 |
| | | | (0.019) | (0.025) | (0.022) | (0.022) |
| F_DumXT_cash | | | | 0.253 | | |
| | | | | (0.563) | | |
| F_DumXT_lev | | | | | 0.092 | |
| | | | | | (0.111) | |

**Table 7.** *Cont.*

| | (1)<br>Delta_Cash | (2)<br>Delta_Cash | (3)<br>Delta_Cash | (4)<br>Delta_Cash | (5)<br>Delta_Cash | (6)<br>Delta_Cash |
|---|---|---|---|---|---|---|
| F_DumXT_PBDITA | | | | | | −0.019 |
| | | | | | | (0.126) |
| _cons | −0.009 | 0.016 | 0.03 | 0.03 | 0.031 | 0.03 |
| | (0.011) | (0.018) | (0.02) | (0.02) | (0.02) | (0.02) |
| Observations | 92 | 83 | 83 | 83 | 83 | 83 |
| R-squared | 0.168 | 0.263 | 0.293 | 0.295 | 0.3 | 0.294 |

Standard errors in parentheses. ***, **, * denotes rejection of null hypothesis at 1%, 5% and 10% significance levels respectively.

Pre-merger holdings of domestic and foreign promoters of the target also negatively affect Delta Cash. This is evident from the findings presented in column 3 of Table 7, wherein Tar_Prom and F_Dummy have an estimated significant coefficient of −0.001 and −0.033, respectively. Acquisitions result in a change of control or dilution of control for target shareholders. When controlling shareholders also enjoy private benefits of control, such a change means giving up such private benefits. Therefore, acquirers have to spend more on post-integration measures.

Post-acquisition cash utilization of the target's cash reserves positively affects the operating expenses of the acquirer. Such an increase in operating expenses is not commensurate with an increase in asset utilization. This was evident from the higher beta coefficients of Delta OER, which were statistically significant, compared to the lower beta coefficient of Delta ATO, which were not significant.

## 6. Conclusions

This paper provides valuable insights into the impacts created by target companies on the changes in agency cost of acquirers. The findings indicate that several target characteristics, such as target ownership and target financials, affect changes in the agency cost of acquirers. The analysis suggests that post-acquisition changes in agency cost of acquirers are significantly different due to the presence of domestic promoters and foreign promoters of target companies. Further, these promoter groups also have conflicting interests, exacerbating the post-acquisition agency cost. Domestic promoters of the target exacerbate the post-acquisition agency cost of the acquirer. However, the monitoring role exercised by foreign promoters plays a significant role in lowering the post-acquisition agency cost of acquirers. These functions are often exercised by disciplining the target's management through debt holdings and monitoring cash reserves. Foreign promoters also favorably affect post-acquisition profitability by negatively affecting post-acquisition operating expenses, thereby indicating that they mitigate the agency cost by monitoring debt, cash, and profitability.

Post-acquisition cash utilization of the target's cash reserves positively affects the operating expenses of the acquirer. However, such an increase in operating expenses is not commensurate with an increase in asset utilization. Furthermore, the pre-merger relative size of the target plays a key role in affecting post-acquisition agency cost. The results indicate that the acquisition of larger targets exacerbates agency costs.

The analysis presented in this study has certain limitations and results should be interpreted cautiously. One limitation is that not all changes in ATO and OER can be attributed to acquisition, as, after all, these accounting metrics are proxies and may not be able to capture the entire gamut of agency conflicts.

The main findings of the study are that the presence of controlling domestic promoters in target companies can have a profound impact on the post-acquisition agency cost of acquirers. The implication of these findings would be significant for an analyst covering M&As, regulators, and policymakers. The outcome of a merger could be gauged (to some extent) based on the degree of change in control for target promoters (domestic and foreign). The study's findings can facilitate policy formulation that takes into consideration the

protection of shareholders after an acquisition. Moreover, regulators can benefit from this study by formulating policies that mandate better disclosure pertaining to post-acquisition changes in cash and debt.

**Author Contributions:** Conceptualization, P.N. and A.K.G.; methodology, P.N. and A.K.G.; software, P.N.; formal analysis, P.N.; investigation, A.K.G.; data curation, P.N.; writing—P.N.; writing—review and editing, A.K.G.; visualization, P.N.; supervision, A.K.G. All authors have read and agreed to the published version of the manuscript.

**Funding:** This research received no external funding.

**Data Availability Statement:** The data presented in this study are available on request.

**Conflicts of Interest:** The authors declare no conflicts of interest.

## Notes

[1] Netter et al. (2011) evidence that the majority of takeover papers analyze less than 5% of acquisitions made by domestic acquirers.

[2] About 85% of all incorporated businesses in India are family businesses (The Economic Times 2023).

[3] Corporate Governance in the Wake of Financial Crisis: Selected International views, 2010, UNCTAD (United Nations 2010).

[4] The results of the regression analysis with the winsorized dataset are similar to the analysis reported in the paper and for the sake of brevity, the regression results with the winsorized dataset are not reported.

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
