# Peer review of "Post-Acquisition Changes in Agency Cost of Acquirers: Effect of Target Companies"

_jrfm, doi:10.3390/jrfm17010011_

Round 1
Reviewer 1 Report
Comments and Suggestions for Authors
The topic of the paper is interesting, and the empirical part is well-written.
However in my opinion the manuscript does not meet the criteria as regards empirical works.
The structure of the paper is not appropriate; the literature review and discussion section should be included in the paper. There is no information if similar research was conducted before. Does this investigation provide something new as regards agency theory (or other theories), asymmetry information, and the M&A process? What is the originality/value of this study?
The research problem and research hypothesis are not explained in the paper. I would recommend to explain the motivation for this research.
In conclusion, the Authors write:
“The implications of this study are significant for regulators such as SEBI, which gov-53 erns the M & A space through the SEBI (Substantial Acquisition of Shares and Takeovers) 54 Regulations, 2011.”
I would recommend explaining if the implications of this study are valid in the international context.
Author Response
Redressal of review comments is presented in the document attached.

Reviewer 2 Report
Comments and Suggestions for Authors
The paper investigates the impact of target company's promotors, financial condition on the post acquisition 'agency' cost for the acquirers. A sample of 91 non-banking and finance companies from India is analysed using fixed effect panel regression models. The two proxy measures of the agency cost are used as dependent variables for acquirers. These are regressed against target companies' leverage, relative size, cash position and few other variables.
The paper has reported a mixed evidence arguing in the end that proxy of asset turnover ratio (ATO) does not help much with understanding the agency cost while operating expense ratio of acquirer has better fit with independent variables.
Overall design of the research is fine. The concept of agency could be explained in more detail context in this case. As the target company's shareholders retain controlling stake it appears in the study, in which case is the agency problem framed between shareholders of acquiring company and its managers? This could be clarified.
The OER changes in the descriptive statistics table 2 need explanation and same applies to regression results. These seem very large movements in operating expenses. A 241% reduction and 753% increase? Am I a misreading this. If -.121 = 12.1% mentioned in the text then -2.419 = 241%? If these are correct numbers they beg the question why would acquirer's operating cost vary so much? Consider significant reduction of 200% percent. Depending on the percentage of shares acquired in the target company, this must mean that target companies operating costs are very very low to overall reduce the OER of acquirer. This needs to be explained here. Seems very counter-intuitive. What is median change in OER given the high SD it may be worth looking at the median.
Overall reasonably well argued conclusions based on the analysis however, explanatory power of the models is weak which is not unusual given that post acquisition performance of acquiring firm is not only affected by the acquisition which is difficult to isolate. For example, all the changes to ATO or OER after acquisition cannot be attributed to acquisition. Perhaps relative size measure captures this partially but it may not be sufficient as it is indicated by low R square.
Some references are not used in the text buy they appear in the reference list (Altman, 1984, Arena et al 2022, Atanasov et al 2008 for example).
Author Response
Kindly find the response to reviewers comments in the file attached.

Reviewer 3 Report
Comments and Suggestions for Authors
The authors examine how agency cost of acquirers changes after aquisition. The paper has originality, but it requires some revisions before it can be published:
- The authors claim in the abstract that foreign promoters of target firms have a key role in lowering the agency costs of acquirers after the deal. However, they also acknowledge that buying bigger targets increases these costs. This may lead to some inconsistent findings, so the authors need to be more accurate.
- The authors did not explain the methods they used or the main findings and their economic consequences in the introduction.
- The authors examined the changes in agency costs of acquirers from 2008–09 to 2019–20. They should expand their study to include more cases of mergers and acquisitions and compare their findings.
- The authors measured the agency cost of acquirers using three different indicators. However, they should cite previous studies to justify their selection of these indicators.
Comments on the Quality of English LanguageThe quality is accepted.
Author Response
Review comments are addressed and can be found in the document attached.

Reviewer 4 Report
Comments and Suggestions for Authors
Please see attached.

Please see attachment.
Author Response
Review comments are addressed and can be found in the attached document

Round 2
Reviewer 1 Report
Comments and Suggestions for Authors
I found that the authors have taken a lot of effort to work on the review comments, however I have one minor recommendation. I think that “Results and Discussion” section might include a few sentences referring to the hypotheses (H1-H4).
Author Response
We thank the reviewer for his comments and acknowledgment. We have incorporated the suggestions.
Reviewer 3 Report
Comments and Suggestions for Authors
The author has made some improvements
Comments on the Quality of English Languageok
Author Response
Thanks for the prior comments and acknowledgment.